# The Compound Response Relationship between Hydro-Sedimentary Variations and Dominant Driving Factors: A Case Study of the Huangfuchuan basin

**Jingwei Yao [1,2], Zhanbin Li [1,3,*], Wenyi Yao [2], Peiqing Xiao [2], Pan Zhang [2], Mengyao Xie [1], Jingshu Wang [2] and Shasha Mei [4]**

1 State Key Laboratory of Eco-Hydraulics in Northwest Arid Region of China, Xi'an University of Technology, Xi'an 710048, China; solofromchina@163.com (J.Y.)
2 Key Laboratory of Soil and Water Conservation on the Loess Plateau of Ministry of Water Resources, Yellow River Institute of Hydraulic Research, Zhengzhou 450003, China
3 State Key Laboratory of Soil Erosion and Dryland Farming on the Loess Plateau, Institute of Soil and Water Conservation, Chinese Academy of Sciences and Ministry of Water Resources, Yangling 712100, China
4 School of Management Engineering, Zhengzhou University of Aeronautics, Zhengzhou 450046, China
* Correspondence: zhanbinli@126.com; Tel.: +86-136-0925-9310

**Abstract:** The Huangfuchuan basin is one of the major sources of coarse sediment in the Yellow River and has long been a focal point and challenge for the conservation of soil and water in the Yellow River Basin. In this study, we analyzed the phase differentiation characteristics of water–sediment variations during the flood season in the Huangfuchuan basin using a long-term dataset. We elucidated the complex response relationship between water–sediment variations and meteorological factors and human activities, which is of great significance for revealing the mechanisms of water–sediment variations in the region and establishing a scientific water–sediment regulation system in the basin. Statistical methods such as the Mann–Kendall trend test, Sen's slope estimation, Pettitt nonparametric test, and principal component analysis were employed to identify and analyze the trends and dominant driving factors before and after the water–sediment variations and abrupt changes in parameters such as rainfall and temperature in the Huangfuchuan basin. Additionally, multiple regression analysis was used to determine the extent of the contribution of climate and human activities to water–sediment variations in the Huangfuchuan basin. The study revealed that the year 2000 was a turning point for water–sediment variations, with decreases of 11.3%, 76.7%, and 85.1% in flood season rainfall, flood season runoff, and flood season sediment transport, respectively. Despite significant changes in the underlying surface conditions of the Huangfuchuan basin, the relationship between flood season sediment transport and flood season runoff remained a power–law relationship. In the absence of obvious abrupt changes in temperature, rainfall, and other meteorological factors, the changes in the underlying surface caused by human activities are the main cause of the changes in runoff and sediment yield in the Huangfuchuan basin. The current level of vegetation restoration in the Huangfuchuan basin is still relatively low, making it difficult to exert stronger control on sediment yield during the flood season. Meanwhile, human activities, primarily based on engineering measures, play a more significant role in the control of soil and water loss in the basin.

**Keywords:** variations in water and sediment; hydrosedimentary relationship; underlying surface; human activities; Huangfuchuan basin

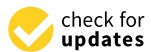



## 1. Introduction

Climate change and human activities can cause changes in river runoff and sediment transport [1–5]. Climate warming accelerates the water cycle in the watershed, leading to changes in precipitation frequency and intensity, which in turn affect surface hydrological

processes [6,7]; the impact of human activities on the watershed's water cycle is mainly manifested in land use/cover change [8], changes in watershed underlying surface conditions caused by the construction of large-scale water conservancy projects, etc., which, in turn, affect the production and accumulation processes within the watershed.

The Huangfuchuan basin is a concentrated source of coarse sediment in the middle reaches of the Yellow River [9]. The basin is extremely fragile in terms of ecology and suffers from severe soil erosion. It has always been a crucial and difficult point for soil and water conservation in the Yellow River Basin. Under the dual impact of climate change and human activities, the Huangfuchuan basin has experienced a dramatic reduction in water and sediment, and the water–sediment relationship has undergone critical changes, attracting widespread attention [10–15]. In the 1950s, the Huangfuchuan basin annually transported $0.64 \times 10^8$ t of sediment to the Yellow River [16], and severe soil erosion resulted in prominent contradictions between humans and the environment [17]. Since the 1960s, a series of soil and water conservation measures have been implemented in the Huangfuchuan basin, including the construction of terraces, check dams, and reservoirs, as well as afforestation and returning farmland to forest, which has led to a decrease in runoff and sediment transport within the basin [18,19]. After the year 2000, with the implementation of large-scale projects such as returning farmland to forests and grasslands, the advancement of check dam construction, the prohibition of deforestation, and the transfer of surplus rural labor, the water–sediment relationship within the basin was profoundly affected [20]. Furthermore, there have been significant changes in the water–sediment conditions in the Yellow River Basin [21–24]. From 2000 to 2020, the vegetation coverage in 97.7% of the Loess Plateau showed an increasing trend, and the water retention capacity increased by 10–30 mm compared to previous years, with each hectare of land retaining 1–5 t of soil. The reduction in sediment caused by soil and water conservation measures in the Huangfuchuan basin accounted for 1.6% of the decrease in sediment deposition in the lower reaches of the Yellow River [25], indicating an important change in the water–sediment situation in the lower reaches of the Yellow River. In addition, both the flood season runoff and sediment transport in the Huangfuchuan basin have shown a significant decreasing trend, but the degree of reduction is different, with the decrease in runoff often being smaller than the decrease in sediment transport [10,24,26,27]. Since the beginning of the 21st century, the interannual variability in water and sediment has become more significant [17]. Studies have shown [26] that compared to the period from 1954 to 1969, the Huangfuchuan basin experienced an 82.6% reduction in runoff and an 84.2% reduction in sediment from 2000 to 2009.

The majority of current research on the attribution analysis of water–sediment variations in watersheds primarily focuses on climate change and human activities. The analytical methods employed include hydrological modeling, the dual cumulative curve method, the cumulative slope method, and the elasticity coefficient method, among others. Currently, the investigation of the causal factors underlying hydrological processes within watersheds is still in its nascent stage, with the mechanisms and synergistic effects of influencing factors yet to be clearly elucidated. Furthermore, disentangling the specific roles of individual factors from the multitude of coacting elements remains a formidable challenge. The changes in water and sediment in the Huangfuchuan basin are the result of the coupled effects of climate factors and human activities, and in recent years, with human activities taking the dominant position, human activities have become the main factor driving the changes in water and sediment in the Huangfuchuan basin [27,28]. Many studies have shown that there was a sudden change in runoff in the Huangfuchuan basin in approximately 1999 [11,17,29–31]. Before 1999, climate change contributed to 64~76% of the runoff reduction; afterwards, the contribution of human activities reached 71~88%. Xu et al. also found in their research [11,31] that precipitation was crucial in the changes in water and sediment in the Huangfuchuan basin before the 21st century, and after entering the 21st century, human activities such as soil and water conservation measures accounted for

approximately 65~70% of the changes in water and sediment, while precipitation accounted for 30~35%.

Currently, most studies have focused on the analysis of the water–sediment relationship using the annual water and sediment data of the Huangfuchuan basin. However, it should be noted that the Huangfuchuan basin, located in the loess hilly and gully region, experiences the majority of its runoff and sediment production during the flood season, which typically occurs from June to September. During this period, rainfall can account for over 80% of the entire flood season's precipitation, with the sediment generated during the flood period constituting more than 85% of the annual sediment load [32]. Therefore, the flood season serves as a crucial juncture for the management of soil erosion in this basin.

Simultaneously, it is imperative to recognize that the variations in water and sediment are influenced by multiple factors, making the analysis of their dynamics highly intricate. To gain a comprehensive understanding of the water–sediment relationship and the mechanisms driving their changes, it is necessary to eliminate the impacts of nonflood phases. In light of this, the present study employs various analytical methods, including the Mann–Kendall trend test, Sen's slope estimation, Pettitt nonparametric test, and principal component analysis. By utilizing long-term precipitation and water–sediment data, this study examines the phase differentiation characteristics of the dominant driving factors behind the flood season's water–sediment changes in the Huangfuchuan basin over the past 66 years. The findings confirm that alterations in underlying surface conditions have not affected the established runoff-sediment transport function in the basin. Furthermore, the study reveals the coupling effects of vegetation restoration and human activities on the flood season's water–sediment changes. The ultimate aim of this research is to provide theoretical foundations and data support for comprehensive soil and water conservation measures in the watershed and the establishment of a scientifically informed water–sediment regulation system.

## 2. Data and Methods

### 2.1. Study Area

The Huangfuchuan is in the upper section of the Helong Interval, between the town of Hekou and Longmen District. It serves as a primary tributary of the middle reaches of the Yellow River, originating at the junction of the Ordos Plateau and the Loess Plateau. Flowing through Shagedu Town of Jungar Banner and merging with the Yellow River in Fugu County, Shaanxi Province, the river basin covers an area of 3246 km$^2$, with 3215 km$^2$ affected by soil erosion, accounting for 99.0% of the total basin area [33]. The main channel stretches 137 km (Figure 1).

This river basin belongs to a typical arid and semiarid climate zone, with extensive exposure of Pisha sandstone. Ecological degradation is severe, and sandy areas are widely distributed. The basin falls within the coarse sand region of the Yellow River, experiencing erosion intensity that is rare in China and even in the world. According to data collected from 1954 to 1969, the annual average precipitation in the basin is 431.2 mm, with an annual average runoff of 207 million m$^3$ and an annual average sediment yield of 62 million t. It is one of the major sandy tributaries of the Yellow River.

The governance of the Huangfuchuan basin began in the 1950s, primarily focusing on measures related to forests and vegetation, with few engineering measures such as terracing and silt embankments [34]. By the 1970s, the level of basin governance was only 6.7%, with forest and vegetation measures accounting for 86.3% and engineering measures accounting for 13.7%. In 1983, the basin was officially designated as one of the eight key areas for comprehensive governance in the country, accelerating the governance pace. By the end of 2015, a total of 886 silt embankments had been constructed in the Huangfuchuan basin, including 507 backbone embankments, with a total storage capacity of 493 million m$^3$. The accumulated silt storage capacity reached 352 million m$^3$, resulting in a remaining capacity ratio of 28.6%. Research has shown that from 2006 to 2019, there has been a "double decline" in soil and water erosion area and intensity in the Huangfuchuan

basin [24]. The area affected by soil erosion decreased from 2778.99 km$^2$ to 1598.98 km$^2$, with a significant reduction in the area of severe and extremely severe soil erosion and a noticeable increase in the area of mild and below soil erosion. As of 2019, the basin's soil and water conservation rate stands at 50.74%, indicating a significant improvement in the ecological environment within the basin.

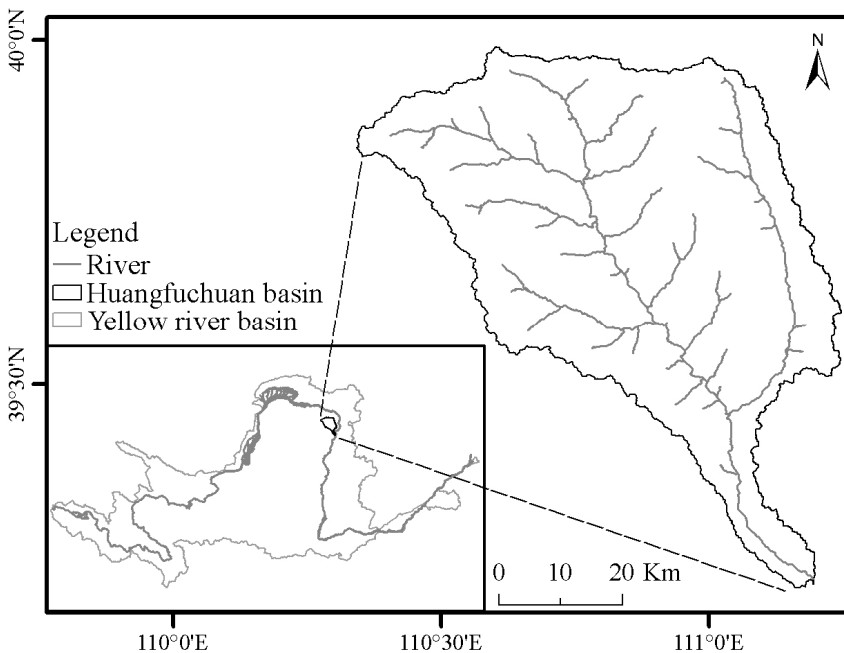

**Figure 1.** Location map of the Huangfuchuan basin.

### 2.2. Data Source and Processing

### 2.2.1. Vegetation Data

Using the MOD13Q1 NDVI product from 2000 to 2019, with a temporal resolution of 16 days and a spatial resolution of 250 m, we selected the annual surface cover dataset (CLCD) for each province in China from 2000 to 2021 [35]. The surface cover product had a temporal resolution of 1 year and a spatial resolution of 30 m, which we resampled to 250 m to match the NDVI product. Based on the 16-day NDVI data, we calculated the average NDVI during the vegetation growing season (from March to November). Combining the surface cover data, we employed pixel dichotomy to convert the NDVI values into vegetation coverage. The formula is as follows:

$$FVC_{i,j} = \frac{NDVI_{i,j} - NDVI_{min}}{NDVI_{max,j} - NDVI_{min}} \tag{1}$$

where $FVC_{i,j}$ is the vegetation coverage of the i-th pixel in the j-th vegetation class. $NDVI_{i,j}$ denotes the average NDVI during the growing season for the i-th pixel in the j-th vegetation class. $NDVI_{max,j}$ corresponds to the 95th percentile value of the annual maximum NDVI histogram for the j-th vegetation class. $NDVI_{min}$ corresponds to the 5th percentile value of the annual minimum NDVI histogram for the bare soil class.

### 2.2.2. Temperature Data

To obtain daily temperature data near the Huangfuchuan basin, three meteorological stations located in Yijinhuoluo Banner, Hequ, and Shenmu were employed. The data spanned 1955 to 2019 (Figure 2, Table 1). Subsequently, an equilateral averaging method was employed to process the temperature data from the three stations.

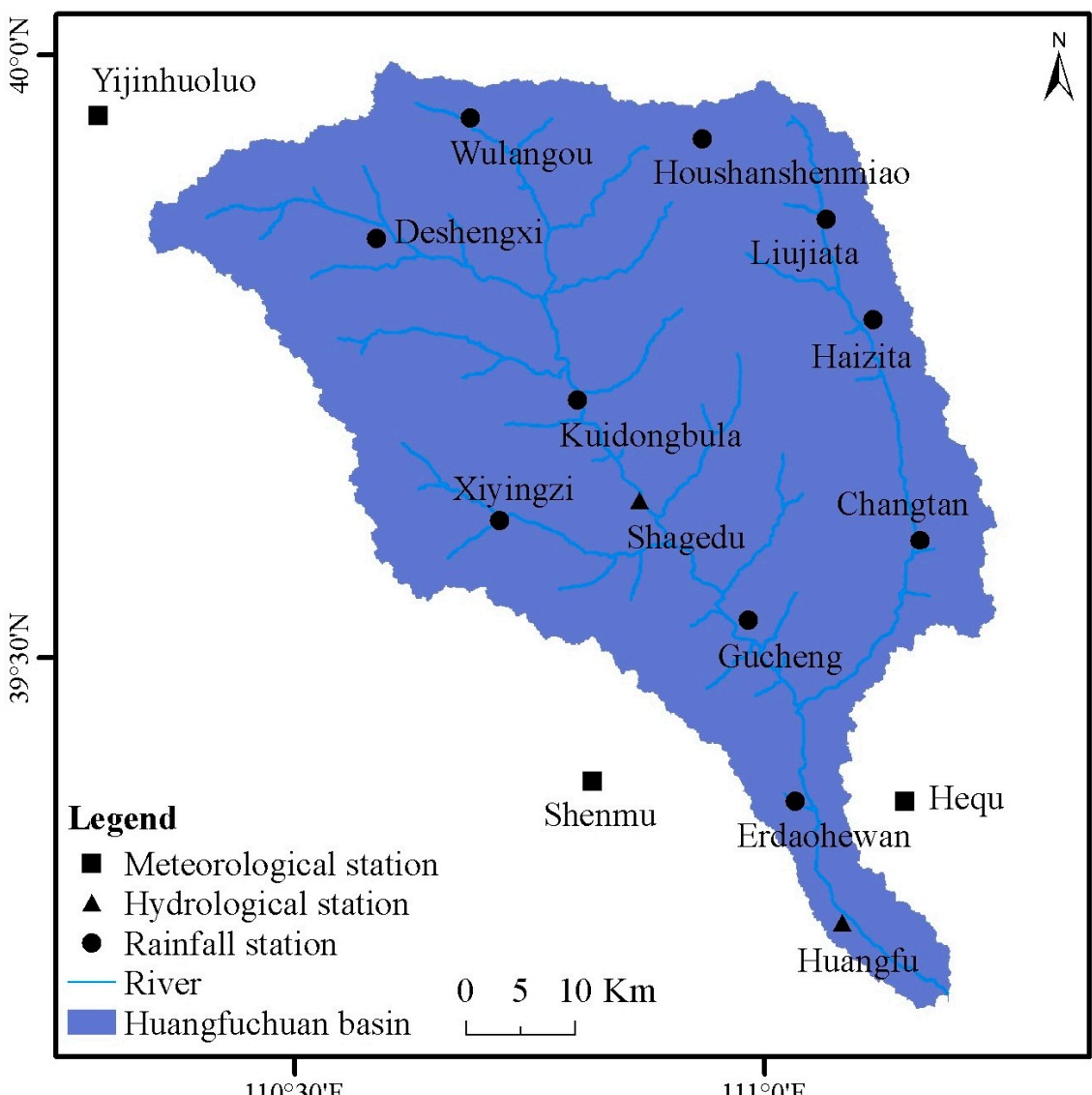

**Figure 2.** Meteorological, precipitation and hydrological stations of the Huangfuchuan basin.

**Table 1.** Weather station information for the Huangfuchuan basin.

| Station Code | Station Name | East Longitude | North Latitude | Altitude (m) | Monitoring Period (Years) |
|---|---|---|---|---|---|
| 53545 | Yijinhuoluo Banner | 109°43′ | 39°34′ | 1330.5 | 1955–2019 |
| 53564 | Hequ | 111°09′ | 39°22′ | 861.5 | 1955–2019 |
| 53651 | Shenmu | 110°25′ | 38°49′ | 941.1 | 1955–2019 |

2.2.3. Precipitation Data

In accordance with the "Hydrological Data of the Yellow River Basin" Volume 3, a careful selection of twelve precipitation stations in the Huangfuchuan basin was undertaken (Figure 2, Table 2). The data covered the period 1955 to 2019. Subsequently, the daily precipitation tables for these twelve stations during the flood season (June to September) were meticulously organised. Utilising the sophisticated Thiessen polygon method in ArcGIS, the average rainfall in the Huangfuchuan basin was precisely calculated.

**Table 2.** Precipitation site of the Huangfuchuan basin.

| Station Code | Station Name | East Longitude | North Latitude | Monitoring Period (Years) |
|---|---|---|---|---|
| 40623350 | Wulangou | 110°41′ | 39°57′ | 1955–2019 |
| 40623500 | Deshengxi | 110°35′ | 39°51′ | 1955–2019 |
| 40623600 | Kuidongbula | 110°48′ | 39°43′ | 1955–2019 |
| 40623650 | Shagedu | 110°52′ | 39°38′ | 1955–2019 |
| 40623700 | Xiyingzi | 110°43′ | 39°37′ | 1955–2019 |
| 40623750 | Gucheng | 110°59′ | 39°32′ | 1955–2019 |
| 40623450 | Houshanshenmiao | 110°56′ | 39°56′ | 1955–2019 |
| 40623800 | Liujiata | 111°04′ | 39°52′ | 1955–2019 |
| 40623900 | Haizita | 111°07′ | 39°47′ | 1955–2019 |
| 40623950 | Changtan | 111°10′ | 39°36′ | 1955–2019 |
| 40624050 | Erdaohewan | 111°02′ | 39°23′ | 1955–2019 |
| 40624100 | Huangfu | 111°05′ | 39°17′ | 1955–2019 |

2.2.4. Runoff and Sediment Data

The compiled streamflow and sediment data from the Huangfu hydrological station in the Huangfuchuan basin were utilised. The data spanned 1955 to 2019, with detailed information regarding the hydrological stations provided in Table 3.

**Table 3.** Hydrographic station of the Huangfuchuan basin.

| Station Code | Station Name | East Longitude | North Latitude | Monitoring Period (Years) | Monitoring Parameters |
|---|---|---|---|---|---|
| 40600900 | Huangfu | 111°05′ | 39°17′ | 1955–2019 | Daily flow<br>Daily sediment<br>transport rate |

2.2.5. Soil and Water Conservation Data

The data on soil and water conservation measures from 1996 to 2019 were sourced from the revised edition of the "Research on Sediment Yield in the Yellow River Basin" compiled by the Yellow River Institute of Hydraulic Research of the Yellow River Conservancy Commission. This comprehensive dataset included information on terracing, afforestation, grass planting, closure-based management, and the area covered by silt retention dams.

*2.3. Research Method*

This study utilizes a suite of sophisticated analytical techniques, namely, the Mann–Kendall trend test [36], Sen's slope estimation nonparametric test [37], Pettitt method [38], and principal component analysis [39], to discern the points of abrupt changes in the water–sediment dynamics within the Huangfuchuan basin over a span of 66 years.

2.3.1. Mann–Kendall Trend Test

The Mann–Kendall trend test is well suited for analysing time series data exhibiting continuous increasing or decreasing trends (monotonic trends). As a nonparametric test, it does not assume a normal distribution of the measurements or require the trend to be linear. Moreover, it is robust against missing values and outliers, causing it to be widely used in trend significance analysis of long-term time series data.

For a time series with a sample size of $X$, the Mann–Kendall statistics yield parameters including the test statistic S and variance $V(S)$.

$$S = \sum_{k=1}^{n-1} \sum_{j=k+1}^{n} \text{sgn}(X_j - X_k) \qquad (2)$$

$$\text{sgn}(X_j - X_k) = \begin{cases} 1 & X_j - X_k > 0 \\ 0 & X_j - X_k = 0 \\ -1 & X_j - X_k < 0 \end{cases} \tag{3}$$

$$V(S) = \frac{1}{18}\left( n(n-1)(2n+5) - \sum_{k=1}^{p} q_k(q_k - 1)(2q_k + 5) \right) \tag{4}$$

where $S$ is the test statistic and $X_j$ and $X_k$ denote the observed values of the corresponding time series for $j$ and $k$, respectively, with $k < j$. $\text{sgn}(\cdot)$ is the sign function, $V(S)$ represents the variance, $n$ is the number of independent and identically distributed samples in the test, $p$ is the number of groups, and $q_k$ represents the number of elements in each group.

To approximate a standard normal distribution for large sample data, the test statistic $S$ is transformed into the $Z_{MK}$ test statistic:

$$Z_{MK} = \begin{cases} \frac{S-1}{V(S)} & S > 0 \\ 0 & S = 0 \\ \frac{S+1}{V(S)} & S < 0 \end{cases} \tag{5}$$

The trend is examined using the $Z_{MK}$ value. A positive (negative) value of $Z_{MK}$ indicates an upwards (downwards) trend in the series being tested. In the Mann–Kendall test, the significance levels used are 0.001, 0.01, 0.05, and 0.1.

### 2.3.2. Sen's Slope Estimation Method

Sen's slope estimation method is employed to calculate the slope of the series, denoted as $\beta$. The slope $\beta$ represents the average rate of change and trend of the time series. When $\beta > 0$, the series exhibits an upwards trend. When $\beta = 0$, the trend of the series is not significant. When $\beta < 0$, the series shows a downwards trend. The calculation formula for Sen's slope of a time series $x_t = (x_1, x_2, \ldots, x_n)$ is as follows:

$$\beta = M_f\left( \frac{x_j - x_i}{j - i} \right), \forall j > i \tag{6}$$

where $M_f$ is the median function.

### 2.3.3. Pettitt Change-Point Test

The Pettitt change-point test is used to determine whether there is a significant change point in a hydrometeorological time series, even when the exact timing of the change is unknown. For a hydrometeorological time series $X = (x_1, \ldots, x_n)$, assuming the change point occurs at $X_t$, the original time series can be divided into two parts: $x_1, x_2 \ldots, x_t$ and $x_{t+1}, x_{t+2}, \ldots, x_n$. The statistic $U_{t,n}$ is defined to assess the possible occurrence of a change point at time t:

$$U_{t,n} = U_{t-1,n} + \sum_{j=1}^{n} \text{sgn}(x_t - x_j) \ t = 2, \cdots, n \tag{7}$$

$U_{i,n} = \sum_{j=1}^{n} \text{sgn}(x_i - x_j)$ and $\text{sgn}(\cdot)$ denote the sign function, which is calculated according to Equation (3).

To determine the probable occurrence time, $t$, of a mutation point, the statistical measure $K_t$ is defined to locate the most likely mutation point.

$$K_t = \max_{1 \le t \le n} |U_{t,n}| \tag{8}$$

After identifying the mutation point using Equation (3), the significance level $P_t$ is calculated using the following formula:

$$P_t = 2\exp\left(\frac{-6K_t^2}{n^3 + n^2}\right) \tag{9}$$

For a given confidence level $\alpha$, if $P_t > \alpha$, the null hypothesis is accepted, indicating no significant mutation at time $t$; if $P_t < \alpha$, the null hypothesis is rejected, indicating a significant mutation at time $t$. In this paper, a confidence level of $\alpha = 0.5$ was chosen.

### 2.3.4. Principal Component Analysis

Principal component analysis is a method of linear dimensionality reduction that allows for the synthesis of multiple original variables into one or several composite indicators, known as principal components, while minimizing the loss of information. Each principal component represents a unique linear combination of $P$ random variables, denoted as $X_1$, $X_2$,..., $X_P$ and is primarily determined by the covariance matrix $A$ (or the correlation matrix $P$) of these variables. Let $n$-dimensional vector $w$ be a mapping vector in the low-dimensional projection space. The formula for maximizing the variance after data mapping is as follows:

$$\max_w \frac{1}{m-1} \sum_{i=1}^{m} w^T(x_i - \overline{x})^2 \tag{10}$$

where $m$ represents the number of data points involved in the dimensionality reduction process, $x_i$ denotes the specific vector expression of random data $i$, and $\overline{x}$ is the mean vector of all the data involved in the reduction.

Assuming that $w$ is a matrix composed of column vectors containing all the feature mapping vectors, this matrix can effectively preserve the information within the data. By subjecting this matrix to algebraic linear transformations, an optimized objective function can be obtained as follows:

$$\min_w tr\left(W^T A W\right), \text{s.t.} W^T W = I \tag{11}$$

where $tr$ denotes the trace of the matrix, while $A$ represents the covariance matrix, which can be expressed as follows:

$$A = \frac{1}{m-1} \sum_{i=1}^{m} (x_i - \overline{x})(x_i - \overline{x})^T \tag{12}$$

The output of PCA is $Y = W'X$, obtained by selecting the top $k$ eigenvectors corresponding to the largest eigenvalues from the covariance matrix. These eigenvectors are then used as column vectors to construct the optimal matrix $W$, ultimately reducing the original dimensionality of $X$ to $k$ dimensions.

Data principal component analysis can be implemented using various analytical tools, such as SPSS Statistics, MATLAB, and R. In this study, we utilized SPSS Statistics software to conduct principal component analysis.

## 3. Results

### 3.1. Variation Characteristics of the Water and Sediment in the Huangfuchuan basin

#### 3.1.1. Trends in Water–sediment Dynamics and Meteorological Elements, and Identification of Mutation Points

Rainfall and temperature were selected as the primary meteorological factors, and the Mann–Kendall trend test method was employed to identify the variations in water–sediment dynamics and meteorological elements during the flood season in the Huangfuchuan basin from 1955 to 2019. Statistical analysis (Table 4) revealed that the $Z_{MK}$ statistics and Sen's slope for both runoff and sediment transport during the flood season

were negative, with significant values at the 0.001 significance level. This indicated a significant decreasing trend in water–sediment dynamics within the Huangfuchuan basin. However, the decreasing trend in rainfall during the flood season was not significant ($p > 0.1$), suggesting the absence of significant change. Furthermore, the $Z_{MK}$ statistics and nonparametric Sen's slope for annual average temperature were positive but not significant, with a significance level of $p > 0.1$, indicating a nonsignificant increase without any abrupt change in annual average temperature.

**Table 4.** Trend analysis of hydrometeorological variables in the Huangfuchuan basin.

| Hydrometeorological Variables | $Z_{MK}$ | Sen's Estimation Quantity | Significance Level ($p$) |
|---|---|---|---|
| Flood season discharge ($10^8$ m$^3$) | −5.94 | −0.027 | 0.001 |
| Flood season sediment transport ($10^8$ t) | −5.72 | −0.008 | 0.001 |
| Flood season precipitation (mm) | −0.60 | −0.567 | >0.1 |
| Annual average temperature(°C) | 0.80 | 0.004 | >0.1 |

Based on the Pettitt test, the examination of abrupt changes in flood discharge, sediment transport, precipitation, and annual average temperature during the flood season indicated that at a significance level of 0.01, a point of abrupt change was detected in flood discharge in 2000, while a point of abrupt change was observed in sediment transport in 2001. However, no significant abrupt changes were identified in either precipitation during the flood season nor annual average temperature.

Based on the results of the abrupt change test, the study period was divided into two periods: Period I, from 1955 to 1999 (prior to the abrupt change point), and Period II, from 2000 to 2019 (after the abrupt change point). Table 5 presents the average values and rates of change for the hydrometeorological variables during these two periods.

**Table 5.** Annual average changes in the key hydrometeorological factors before and after the change point in the Huangfuchuan basin.

| Hydrometeorological Parameters | Pre-Mutation Period (1955–1999) | Post-Mutation Period (2000–2019) | Magnitude of Change | Rate of Change (%) |
|---|---|---|---|---|
| Flood season discharge ($10^8$ m$^3$) | 1.560 | 0.364 | −1.196 | −76.7 |
| Flood season sediment transport ($10^8$ t) | 0.510 | 0.076 | −0.434 | −85.1 |
| Flood season precipitation (mm) | 411.6 | 365.0 | −46.6 | −11.3 |
| Annual average temperature(°C) | 8.2 | 8.9 | 0.7 | 8.5 |

Following the abrupt change point, the precipitation during the flood season decreased by 11.3%, while the temperature increased by 8.5%. However, the flood discharge and sediment transport decreased significantly by 76.7% and 85.1%, respectively. This suggested that the changes in precipitation and temperature did not completely align with the changes in flood discharge and sediment transport in terms of trends and magnitudes. Hence, it can be inferred that the reduction in runoff and sediment was likely more closely linked to large-scale land restoration efforts, such as afforestation, grassland restoration, and the construction of sediment retention dams, along with other soil and water conservation measures.

Before the 1970s, the soil and water conservation level in the Huangfuchuan basin was merely 6.8%. However, by 1989 and 1997, it increased to 17.1% and 28.2%, respectively. The number of sediment retention dams in the Huangfuchuan basin increased from 390 in 1978 to 567 in 2010, covering an area of 2216.47 km$^2$, which accounted for nearly two thirds of the total basin area. Additionally, since the 1980s, coal mining, river sand mining, and the development and utilization of water resources in the area have significantly increased. These land use changes caused by human activities altered the underlying conditions of

the basin to varying degrees, which in turn partially impacted the sharp decline in water and sediment in the basin.

### 3.1.2. The Relationship between Water and Sediment and Precipitation Response

The functional relationship between precipitation and flood discharge during the flood season in the Huangfuchuan basin, before and after the year of abrupt change, is presented in Figure 3.

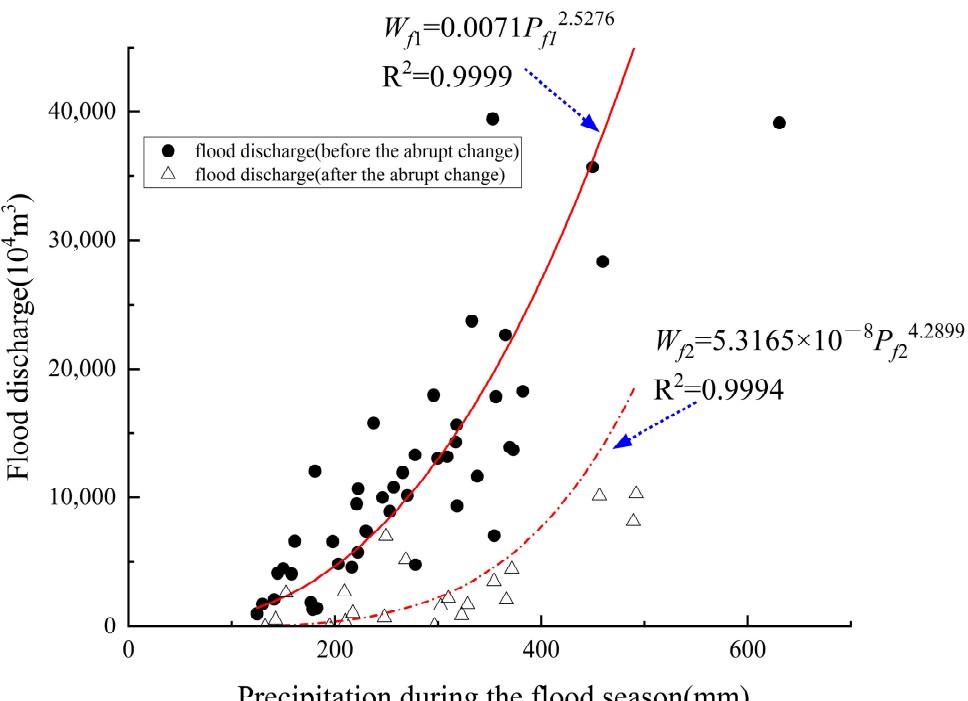

**Figure 3.** The relationship between precipitation during the flood season and flood discharge.

By means of curve fitting, the relationship between precipitation during the flood season and flood discharge before and after the abrupt change is as follows:

$$W_{f1} = 0.0071P_{f1}{}^{2.5276} , \tag{13}$$

$$W_{f2} = 5.3165 \times 10^{-8}P_{f2}{}^{4.2899}. \tag{14}$$

In the equation, $P_{f1}$ and $P_{f2}$ represent the precipitation before and after the mutation, respectively; $W_{f1}$ and $W_{f2}$ represent the runoff before and after the mutation during the flood season. The correlation coefficients $R^2$ of Equations (13) and (14) were 0.9999 and 0.9994, respectively.

Equation (14) indicates that the correlation coefficient between the precipitation and runoff during the flood season slightly decreased after mutation. However, the correlation remained high, suggesting that the magnitude of the post-mutation runoff during the flood season was still primarily determined by precipitation. Nevertheless, it was influenced to a greater extent by anthropogenic factors such as soil and water conservation activities. Furthermore, the functional relationship between the post-mutation runoff during the flood season and the precipitation did not undergo significant changes. Overall, the relationship between the two variables still followed the power-law exponential pattern. For instance, the fitting formula for the upper bound of the relationship between the post-mutation runoff during the flood season and the precipitation remains in the following form:

$$W_{fi} = kP_{fi}{}^{\alpha}. \tag{15}$$

However, the runoff significantly decreased for the same amount of precipitation. For instance, the runoff per unit precipitation was 0.071 thousand m$^3$ before the mutation, whereas after the mutation, it decreased to only $5.3165 \times 10^{-6}$ million m$^3$. This corresponded to a reduction of 97.5% in the runoff per 100 mm of precipitation. The statistical ranges of the flood season precipitation and runoff before the mutation were 124.56–630.80 mm and 9836.9–394,420.3 thousand m$^3$, respectively. After the mutation, the statistical ranges of the flood season precipitation and runoff were 132.36–489.32 mm and 490.0–102,906.5 thousand m$^3$, respectively.

By analysing Equations (13) and (14), it was observed that within the entire range of flood season precipitation, the runoff during the flood season significantly decreased after the mutation for the same level of flood season precipitation. For example, at a flood season precipitation of 370 mm, the runoff before the mutation was 220.11 million m$^3$, whereas after the mutation, it was reduced to 55.33 million m$^3$, representing a decrease of 74.9%. Additionally, before the mutation, the unit runoff during the flood season increased rapidly with an increase in unit flood season precipitation. However, after the mutation, the unit runoff during the flood season increased relatively slowly with an increase in unit flood season precipitation. In other words, for different levels of flood season precipitation, the variation in the flood season runoff after the mutation was significantly smaller than that before the mutation.

Based on the above analysis, it could be inferred that in the absence of any significant changes in meteorological factors such as temperature and precipitation, there has been a marked decrease in the flood season runoff in the Huangfuchuan basin, indicating a mutation. This was likely to be closely related to the changes in underlying surface conditions caused by human activities.

Similarly, the relationship between the flood season precipitation and sediment transport during the flood season in the Huangfuchuan basin was analyzed, and the fitting function is shown in Figure 4.

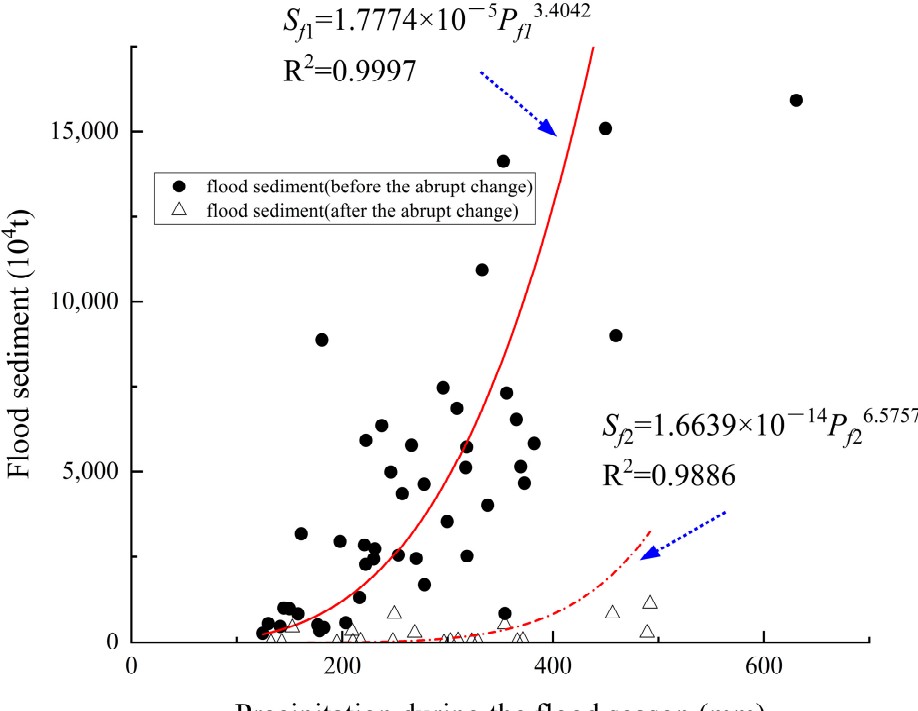

**Figure 4.** The relationship between flood season precipitation and sediment transport during the flood season.

By means of curve fitting, the relationship between flood season precipitation and sediment transport during the flood season before and after the mutation can be described as follows:

$$S_{f1} = 1.7774 \times 10^{-5} P_{f1}{}^{3.0442} , \tag{16}$$

$$S_{f2} = 1.6639 \times 10^{-14} P_{f2}{}^{6.5757}. \tag{17}$$

In the equations, $P_{f1}$ and $P_{f2}$ represent the flood season precipitation before and after the mutation, respectively, while $S_{f1}$ and $S_{f2}$ represent the sediment transport during the flood season before and after the mutation, respectively. The correlation coefficients $R^2$ for Equations (16) and (17) were 0.9997 and 0.9886, respectively. The statistical ranges of the flood season precipitation and sediment transport before the mutation were 124.56–630.80 mm and 2.72 million–159.14 million t, respectively. After the mutation, the statistical ranges of the flood season precipitation and sediment transport were 132.36–492.00 mm and 0.01 million–29.01 million t, respectively.

The relationship between flood season precipitation and sediment transport after the mutation relatively weakened. As depicted in Figure 4, most data points representing the sediment transport during the flood season after the mutation were located below those before the mutation, indicating a decrease in sediment transport for the same level of precipitation. Furthermore, the slope of the trend line became less steep, suggesting a reduced influence of precipitation on sediment transport. For instance, within the entire range of flood season precipitation, the sediment transport during the flood season after the mutation was significantly lower than that before the mutation for the same level of precipitation. Additionally, the increase in sediment transport during the flood season after the mutation was slower with an increase in flood season precipitation. Overall, under the same flood season precipitation, the variation in sediment transport during the flood season after the mutation was considerably smaller than that before the mutation. Similarly, it could be inferred that human activities significantly intensified the impact of changes in underlying surface conditions on sediment production within the basin. Figure 5 shows the fitting function relationship between flood season runoff and sediment transport in the Huangfuchuan basin before and after the mutation.

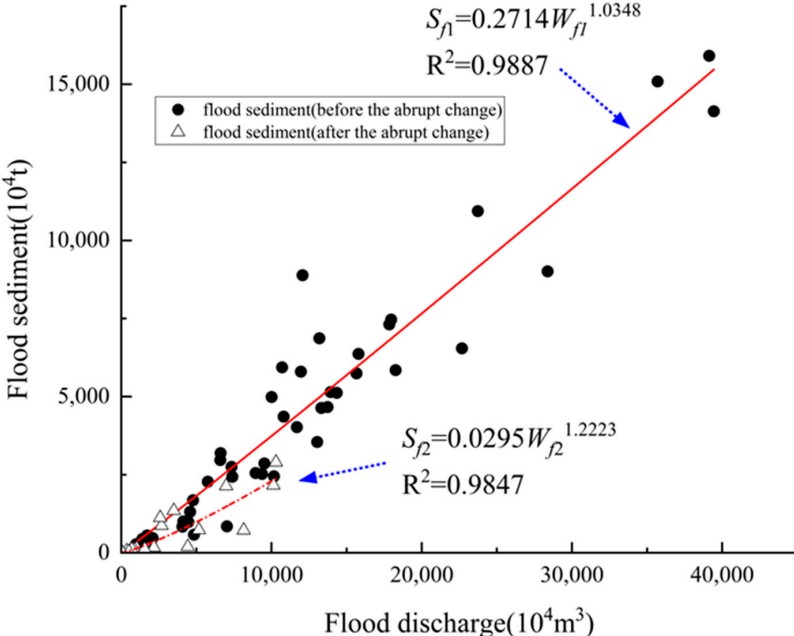

**Figure 5.** The correlation between flood discharge and sediment transport during the flood season.

By means of curve fitting, the relationship between flood discharge and sediment transport during the flood season, both before and after the abrupt change, can be expressed as follows:

$$S_{f1} = 0.2714W_{f1}{}^{1.0348} ,$$ (18)

$$S_{f2} = 0.0295W_{f2}{}^{1.2223}$$ (19)

In the equation, $W_{f1}$ and $W_{f2}$ represent the flood discharge before and after the abrupt change, respectively, while $S_{f1}$ and $S_{f2}$ represent the sediment transport during the flood season before and after the change. The correlation coefficients $R^2$ for Equations (18) and (19) were 0.9887 and 0.9847, respectively.

The statistical ranges for flood discharge and sediment transport before the change were 9.84 million to 394.42 million $m^3$ and 2.72 million to 159.14 million t, respectively. After the change, the statistical ranges for flood discharge and sediment transport during the flood season were 0.49 million to 102.91 million $m^3$ and 0.01 million to 29.01 million t, respectively.

Equations (18) and (19) indicate that both flood discharge and sediment transport during the flood season, before and after the change, exhibited a strong power–law relationship, with an exponent greater than one. This suggested that the flood–sediment relationship remained unchanged before and after the abrupt change. After the change, the average flood discharge in the Huangfuchuan basin was 30.95 million $m^3$, and the average sediment transport during the flood season was 6.23 million t, representing decreases of 74.12% and 86.13%, respectively, compared to that before the change.

In summary, following the abrupt change in flood–sediment dynamics during the flood season, although the runoff and sediment yield per unit rainfall noticeably decreased, the relationship between flood discharge and sediment transport remained consistent, following the same pattern. However, the sediment yield per unit flood discharge significantly decreased by approximately 89.13%, while the influence of flood discharge on sediment transport increased.

### 3.2. The Relationship between Dominant Driving Factors and Their Response

The dominant driving factors of hydrosedimentary changes in the Huangfuchuan basin were identified through principal component analysis. Key factor indicators such as flood season rainfall, sediment transport during the flood season, vegetation cover, and soil conservation were selected.

By calculating the response relationships between the major influencing factors in the Huangfuchuan basin from 2000 to 2019, a comprehensive understanding of the correlation levels between variables could be obtained (Table 6). Table 6 shows that there is a certain degree of correlation between sediment transport during the flood season and factors such as flood season rainfall, vegetation cover, and soil conservation. Specifically, sediment transport during the flood season showed a positive correlation with flood season rainfall while exhibiting a negative correlation with vegetation cover and soil conservation. There was a stronger relationship between sediment transport during the flood season and soil conservation, while the relationship with vegetation cover was comparatively weaker. This indicated that comprehensive soil conservation measures, such as terrace construction, afforestation, and the construction of silt detention dams, had a significant impact on reducing sediment transport during the flood season in the Huangfuchuan basin.

Principal components were determined by calculating the variances as percentages using eigenvalues and eigenvectors. The criteria for selecting the principal components were either eigenvalues greater than one or cumulative variance percentages greater than 80%. Additionally, the eigenvalues of the principal components were calculated (Table 7). By examining the scree plot of the eigenvalues for each component (Figure 6), it was evident that the first two components have the highest slopes and encompassed 90.75% of the information from the original variables.

**Table 6.** Correlation coefficient matrix of key factor indicators in the Huangfuchuan basin.

| Indicator | Precipitation during the Flood Season | Sediment Transport during the Flood Season | Vegetation Coverage | Degree of Water and Soil Conservation Measures |
|---|---|---|---|---|
| Precipitation during the flood season | 1.00 | 0.45 *** | 0.33 *** | 0.18 *** |
| Sediment transport during the flood season | 0.45 *** | 1.00 | −0.34 *** | −0.48 *** |
| Vegetation coverage | 0.33 *** | −0.34 *** | 1.00 | 0.89 ** |
| Degree of water and soil conservation measures | 0.18 *** | −0.48 *** | 0.89 ** | 1.00 |

Note: The symbols *** and ** represent the significance levels of 1% and 5%, respectively.

**Table 7.** Explanation of total variance for each component.

| Components | Initial Characteristic Values | | | Extraction of Squared Sum of Loadings | | |
|---|---|---|---|---|---|---|
| | Eigenvalues | Variance Percentage | Cumulative Percentage | Eigenvalues | Variance Percentage | Cumulative Percentage |
| $Z_1$ | 2.19 | 54.83 | 54.83 | 2.19 | 54.83 | 54.83 |
| $Z_2$ | 1.44 | 35.92 | 90.75 | 1.44 | 35.92 | 90.75 |
| $Z_3$ | 0.27 | 6.79 | 97.54 | | | |
| $Z_4$ | 0.10 | 2.46 | 100.00 | | | |

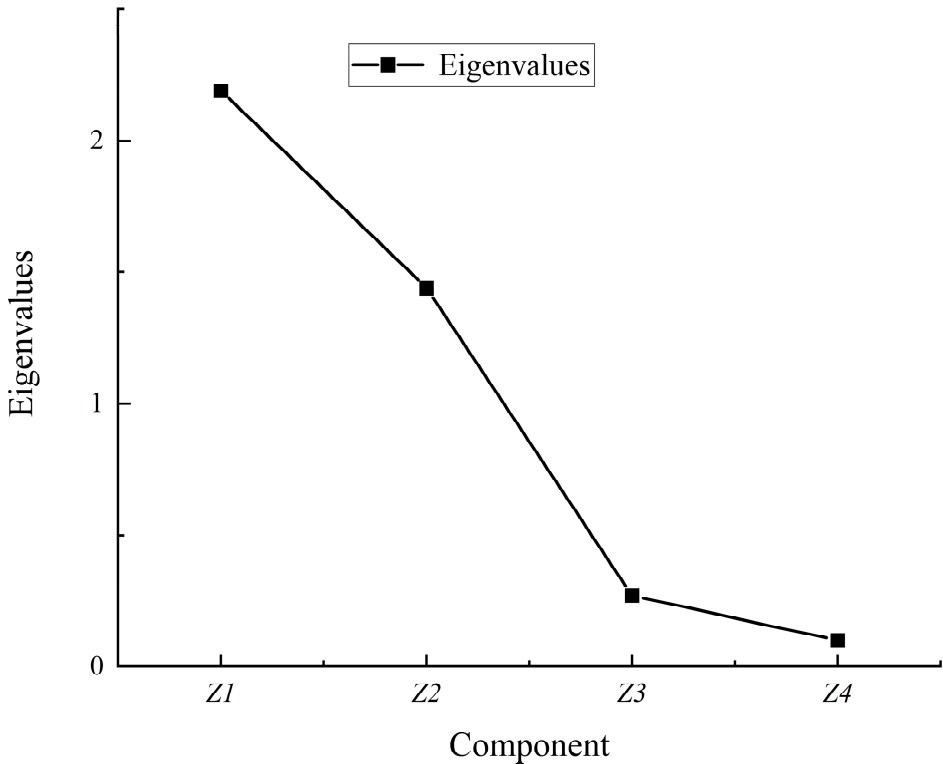

**Figure 6.** Principal Component Analysis (PCA) Scree Plot.

Hence, $Z_1$ and $Z_2$ were identified as the principal components (Table 8).

**Table 8.** Composition of principal component loadings.

| Principal Components | Flood Season Precipitation | Flood Season Sediment Transport | Vegetation Coverage | Degree of Water and Soil Conservation Management |
|---|---|---|---|---|
| $Z_1$ | 0.189 | −0.59 | 0.938 | 0.964 |
| $Z_2$ | 0.928 | 0.723 | 0.226 | - |

Based on the loading analysis for principal components $Z_1$ and $Z_2$, it could be observed that principal component $Z_1$ had significant loadings on soil conservation and vegetation cover, suggesting that it was the primary component influenced by human activities. On the other hand, principal component $Z_2$ exhibited substantial loadings on flood season rainfall, indicating that it was the primary component influenced by natural environmental factors.

Examining the comprehensive scores of the principal components (Table 9, Figure 7), it became evident that the human impact on the hydrosediment changes in the Huangfuchuan basin progressively strengthened after 2007. In particular, since 2013, the influence of human activities has surpassed that of natural environmental factors. Notably, 2012 and 2016 were heavily affected by rainfall. This was supported by the relevant literature, which confirms the occurrence of heavy rainfall and flooding in the Huangfuchuan basin during the flood seasons of these years [40–42]. Specifically, on 20–21 July 2012, there were heavy to torrential rainfalls in the northern part of the Shan-Shaan sector, causing localized downpours and increased water levels in some tributaries. The maximum peak flow at the Huangfu hydrological station on 21 July reached 4700 m$^3$/s, with a sediment transport of 11.49 million t. Similarly, from 17 to 18 August 2016, the northern part of the Shan-Shaan sector experienced heavy to torrential rainfall, with rain in the Huangfuchuan basin lasting approximately 28 h, starting at approximately 4 a.m. on 17 August and ending at approximately 8 a.m. on 18 August. This rainfall covered almost the entire basin, resulting in a maximum peak flow of 2220 m$^3$/s and a maximum sediment concentration of 510 kg/m$^3$. Therefore, the findings of the principal component analysis aligned with the actual situation.

**Table 9.** Comprehensive scores of principal components.

| Year | $Z_1$ | $Z_2$ | Scores | Rank |
|---|---|---|---|---|
| 2016 | 1.061 | 3.101 | 2.031 | 1 |
| 2018 | 1.945 | 0.772 | 1.386 | 2 |
| 2019 | 2.072 | 0.391 | 1.272 | 3 |
| 2013 | 1.237 | 0.951 | 1.097 | 4 |
| 2012 | −0.894 | 3.269 | 1.085 | 5 |
| 2017 | 1.961 | −0.012 | 1.025 | 6 |
| 2014 | 1.703 | 0.234 | 1.002 | 7 |
| 2008 | 0.376 | −0.222 | 0.088 | 8 |
| 2015 | 1.279 | −1.298 | 0.049 | 9 |
| 2010 | 0.624 | −0.86 | −0.083 | 10 |
| 2007 | 0.245 | −0.799 | −0.252 | 11 |
| 2009 | 0.471 | −1.211 | −0.328 | 12 |
| 2004 | −1.182 | 0.257 | −0.496 | 13 |
| 2006 | −1.909 | 0.87 | −0.583 | 14 |
| 2003 | −3.177 | 1.976 | −0.724 | 15 |
| 2005 | −0.243 | −1.434 | −0.805 | 16 |
| 2011 | 0.384 | −2.181 | −0.836 | 17 |
| 2002 | −1.410 | −1.103 | −1.263 | 18 |
| 2001 | −2.432 | −0.783 | −1.645 | 19 |
| 2000 | −2.116 | −1.917 | −2.021 | 20 |

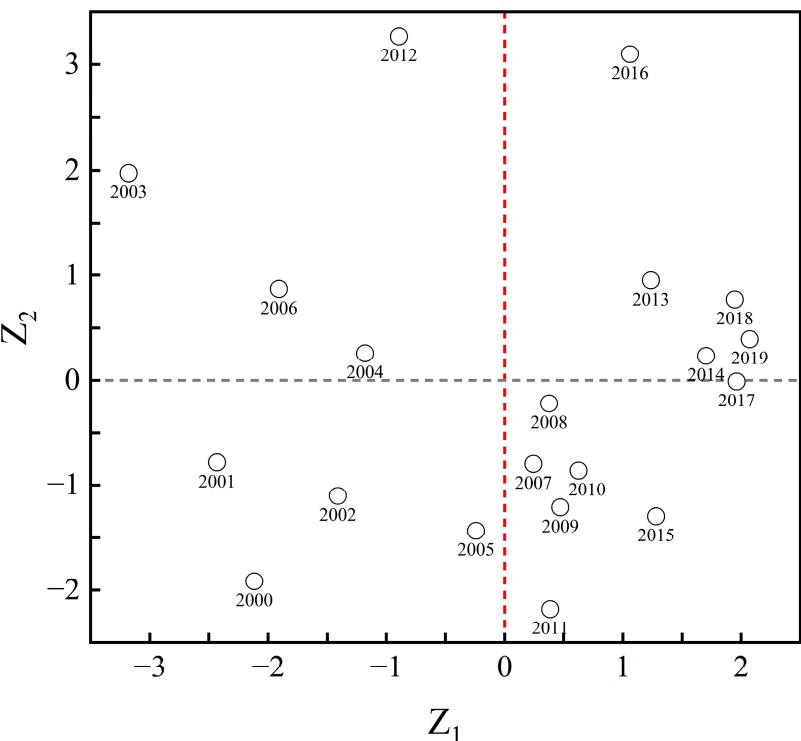

**Figure 7.** Principal component analysis (PCA) score plot.

Based on the correlation matrix and principal component analysis results, there existed a strong correlation between flood season discharge, vegetation cover, and soil conservation measures with sediment transport during the flood season. To establish the complex relationship between hydrosediment changes in the Huangfuchuan basin and the dominant driving factors, a multiple regression analysis was conducted using IBM SPSS Statistics 26 software. This analysis aimed to statistically analyze the composite relationship of sediment transport during the flood season in the Huangfuchuan basin. As a result of the fitting process, the following composite relationship for sediment transport during the flood season was derived:

$$S_f = 4.7103\,P_f - 5.6512\,C_f - 42.6873\,K_f + 792.6563,\ \mathrm{R}^2 = 0.6271. \tag{20}$$

In the equation, $S_f$ represents the sediment transport during the flood season in million t, $P_f$ denotes the flood season precipitation in million cubicm, $C_f$ represents the percentage of vegetation cover, and $K_f$ indicates the percentage of soil conservation measures. The coefficient of determination for Equation (20) was 0.6271. Equation (20) revealed that flood season precipitation positively correlated with sediment transport, while vegetation cover and soil conservation measures negatively correlated with sediment transport. Furthermore, the integrated soil conservation measures exhibited the strongest impact in reducing sediment transport during the flood season.

## 4. Discussion

Under natural conditions, precipitation is the predominant climatic factor influencing water and sediment dynamics. However, in recent years, the implementation of national strategies such as ecological civilization, ecological conservation in the Yellow River Basin, and high-quality development, particularly the conversion of cultivated land into forests and grasslands through projects such as afforestation and reforestation, has shifted the focus of water and sediment changes in the Huangfuchuan basin towards human activities such as soil and water conservation. The extensive transformation of farmland into forests and grasslands in the Huangfuchuan basin has increased vegetation coverage on the surface,

altering the surface conditions and leading to significant vegetation recovery. This has enhanced the ability of vegetation in the basin to intercept and infiltrate water and to cause the water to evaporate, thereby weakening the dynamic relationship between water and sediment. The increased vegetation coverage effectively intercepted and retained runoff sediment, while the vegetation canopy reduced the volume and intensity of rainfall reaching the ground [43], thus mitigating the erosive impact of rainfall on the surface. Moreover, the increase in soil available nutrients significantly promoted the formation of soil aggregates [11], which contributed to water retention and benefitted ecological restoration and plant growth [44,45]. These changes clearly influenced the variations in runoff and sediment in the region.

Furthermore, in recent years, the construction of sediment retention dams and the large-scale transfer of surplus labor from rural to urban areas in the Huangfuchuan basin greatly reduced disturbances and damage to the basin surface. These factors critically impacted the changes in the water and sediment dynamics. The analysis presented in this paper fully demonstrates that the dominant driving factors behind the changes in water and sediment dynamics in the Huangfuchuan basin shifted from natural factors, such as rainfall, to human activities, such as soil and water conservation.

We further confirm that the relationship between runoff and sediment in the Huang-fuchuan basin remains stable. Regardless of the complexity and intensity of climate change and human activities, the statistical relationship between peak runoff and sediment transport during the flood season remains consistent before and after abrupt changes in water and sediment. This finding is consistent with previous research conclusions [22]. In the Huangfuchuan basin and the middle reaches of the Yellow River, which are characterised by high sediment concentrations and coarse sediment, the water and sediment dynamics are influenced by multiple factors resulting from natural geographical conditions. We reveal the complexity of this hydrological process by analyzing the relationships between rainfall, runoff, sediment transport, vegetation coverage, and the degree of soil and water conservation during the flood season. Furthermore, we provide valuable insights into the directionality and variability of the driving factors, which can assist future research in this field.

## 5. Conclusions

(1) The relationship between precipitation, runoff, and sediment transport during the flood season in the Huangfu River Basin follows a power function. The variability of water and sediment is characterized by a threshold relationship with rainfall as the driving factor. Under the same amount of precipitation during the flood season, both runoff and sediment transport exhibit significantly lower values after an abrupt change compared to those before the change. Furthermore, the magnitude of change in runoff and sediment transport after the abrupt change is noticeably smaller.

(2) The variations in water and sediment in the Huangfu River Basin result from the coupling effects of multiple factors, including underlying surface conditions and meteo-rological factors. However, the driving forces of these factors do not exhibit synergistic behavior. Sediment transport during the flood season shows a positive correlation with precipitation while exhibiting a negative correlation with vegetation coverage and soil conservation measures. The continuous improvement in the degree of Huangfuchuan basin soil and water conservation measures has gradually enhanced the control effect on soil erosion during the flood season. However, the extent of vegetation coverage in the basin is relatively low, posing difficulty in significantly reducing sediment transport during the flood season.

(3) The dominant driving factors of water and sediment variations exhibit temporal variations during different historical periods. Prior to the abrupt change in water and sediment conditions, the leading driving factor was flood season rainfall. However, with the intensification of human activities such as large-scale afforestation and check dam con-

struction, soil conservation measures have become the dominant driving force, surpassing the influence of rainfall changes.

(4) Despite significant changes in the underlying surface conditions in the Huangfu River Basin, the functional relationship between sediment transport and runoff during the flood season remains unchanged. In other words, the current underlying surface conditions in the basin do not exert regular regulatory effects on the runoff–sediment relationship.

The processes of runoff generation and sediment transport in a watershed exhibit nonlinearity and uncertainty. They represent energy transformation and material transport processes with interrelationships, states, and characteristics. Therefore, future research needs to be based on systems theory and adopt more scientific and comprehensive theoretical approaches. This will help determine the quantitative relationships and interactions between the various elements of the hydrological system in the Huangfu River Basin, enabling the exploration of simulation and prediction theories and methods for complex watershed systems under strong human-induced disturbances.

**Author Contributions:** Conceptualization, Data curation, Investigation, Methodology, Formal analysis, Writing—original draft: J.Y.; Conceptualization, Methodology, Supervision, Funding acquisition: Z.L.; Formal analysis, Validation, Writing—review and editing: W.Y.; Project administration, Funding acquisition. P.X.; Supervision: P.Z.; Validation, Writing—review and editing: M.X., J.W. and S.M. All authors have read and agreed to the published version of the manuscript.

**Funding:** This research was supported by the National Key Research and Development Program of China (Grant No.: 2022YFF1300805), the National Natural Science Foundation of China (Grant No.: U2243210) and the National Natural Science Foundation of China Major Projects (Grant No.: 42041006).

**Data Availability Statement:** The datasets generated during the current study are available from the corresponding author upon reasonable request.

**Conflicts of Interest:** The authors declare no conflict of interest.

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
