# Peer review of "The Compound Response Relationship between Hydro-Sedimentary Variations and Dominant Driving Factors: A Case Study of the Huangfuchuan basin"

_sustainability, doi:10.3390/su151813632_

Round 1

Reviewer 1 Report

1. Abstract: The summary should be written in between 5 and 8 sentences to ensure that the core content of the experiment is conveyed to the reader. For example:

Sentences 1 and 2: Introduce the research and the reason for choosing the topic (Please note that the novelty of the research is very important).

Sentence 3: What research tools and methods do you use in this article? (I have read through this content in your article, but the content is not written very clearly, the author should consider rewriting this content to make it easier for readers to grasp the information)

Sentences 4 to 7: Main results of the research, write 1 sentence for each result.

Sentence 8: Conclusion and recommendations (if any).

2. Introduction: This part needs to be improved. The Introduction section should briefly place the study in a broad context and highlight why it is important. It should define the purpose of the work and its significance. The current state of the research field should be reviewed, and recent key publications should be cited. Please highlight controversial and diverging hypotheses. Finally, briefly mention the main aim of the work and highlight the principal conclusions. However, the novelty and significance of the manuscript were not highlighted in the Introduction section, please modify the introduction more clearly.

3. Conclusion: Authors should improve this section. Authors should shorten this section with significant conclusions, and revise this section for the better understanding of the topic and its future research.

Author Response

We thank the reviewers for the time and effort that they have put into reviewing the previous version of the manuscript. Their suggestions have enabled us to improve our work. Based on the instructions provided, we uploaded the file of the revised manuscript. Accordingly, we have uploaded a copy of the original manuscript with all the changes highlighted by using the track changes mode in MS Word. Appended to this letter is our point-by-point response to the comments raised by the reviewers. The comments are reproduced and our responses are given directly afterward in a different color (red). 

Responses to the comments of Reviewer 1

  1. Abstract: The summary should be written in between 5 and 8 sentences to ensure that the core content of the experiment is conveyed to the reader. For example:Sentences 1 and 2: Introduce the research and the reason for choosing the topic (Please note that the novelty of the research is very important).Sentence 3: What research tools and methods do you use in this article? (I have read through this content in your article, but the content is not written very clearly, the author should consider rewriting this content to make it easier for readers to grasp the information)Sentences 4 to 7: Main results of the research, write 1 sentence for each result.Sentence 8: Conclusion and recommendations (if any).

Response: Thank you for the abstract suggested. We've taken your advice completely.The precedent version of the abstract has been replaced.Please check Line 15-38.

Sentences 1 and 2 :Introduce the research and the reason for choosing the topic, especially clarified the novelty of the research.we changed this part to “The Huangfuchuan Basin is one of the major sources of coarse sediment in the Yellow River and has long been a focal point and challenge for the conservation of soil and water in the Yellow River basin. In this study, we analyzed the phase differentiation characteristics of water-sediment variations during the flood season in the Huangfuchuan Basin using a long-term dataset. We elucidated the complex response relationship between water-sediment variations and meteorological factors and human activities, which is of great significance for revealing the mechanisms of water-sediment variations in the region and establishing a scientific water-sediment regulation system in the basin. ”

Sentence 3: Introduce the research tools and methods,and we also rewriting this content to make it easier for readers to grasp the information.we changed this part to “Statistical methods such as the Mann–Kendall trend test, Sen’s slope estimation, Pettitt nonparametric test, and principal component analysis were employed to identify and analyze the trends and dominant driving factors before and after the water-sediment variations and abrupt changes in parameters such as rainfall and temperature in the Huangfuchuan Basin.”

Sentences 4 to 7: Main results of the research, write 1 sentence for each result.we changed this part to “Additionally, multiple regression analysis was used to determine the extent of the contribution of climate and human activities to water-sediment variations in the Huangfuchuan Basin. The study revealed that the year 2000 was a turning point for water-sediment variations, with decreases of 11.3%, 76.7%, and 85.1% in flood season rainfall, flood season runoff, and flood season sediment transport, respectively. Despite significant changes in the underlying surface conditions of the Huangfuchuan Basin, the relationship between flood season sediment transport and flood season runoff remained a power-law relationship. In the absence of obvious abrupt changes in temperature, rainfall, and other meteorological factors, the changes in the underlying surface caused by human activities are the main cause of the changes in runoff and sediment yield in the Huangfuchuan Basin. ”

Sentence 8: Conclusion and recommendations (if any).we changed this part to “The current level of vegetation restoration in the Huangfuchuan basin is still relatively low, making it difficult to exert stronger control on sediment yield during the flood season, Meanwhile, human activities, primarily based on engineering measures, play a more significant role in the control of soil and water loss in the basin.”

  1. Introduction: This part needs to be improved. The Introduction section should briefly place the study in a broad context and highlight why it is important. It should define the purpose of the work and its significance. The current state of the research field should be reviewed, and recent key publications should be cited. Please highlight controversial and diverging hypotheses. Finally, briefly mention the main aim of the work and highlight the principal conclusions. However, the novelty and significance of the manuscript were not highlighted in the Introduction section, please modify the introduction more clearly.

Response:Thank you for the Introduction suggested. We've taken your advice completely.we added some sentences to highlight the novelty and significance of the manuscript in the Introduction section, and modify the introduction more clearly.Please check Line 101-125, page 2.we changed this part to “Currently, most studies have focused on the analysis of the water-sediment relationship using the annual water and sediment data of the Huangfuchuan Basin. However, it should be noted that the Huangfuchuan Basin, located in the loess hilly and gully region, experiences the majority of its runoff and sediment production during the flood season, which typically occurs from June to September. During this period, rainfall can account for over 80% of the entire flood season’s precipitation, with the sediment generated during the flood period constituting more than 85% of the annual sediment load. Therefore, the flood season serves as a crucial juncture for the management of soil erosion in this basin.Simultaneously, it is imperative to recognize that the variations in water and sediment are influenced by multiple factors, making the analysis of their dynamics highly intricate. To gain a comprehensive understanding of the water-sediment relationship and the mechanisms driving their changes, it is necessary to eliminate the impacts of nonflood phases. In light of this, the present study employs various analytical methods, including the Mann–Kendall trend test, Sen’s slope estimation, Pettitt nonparametric test, and principal component analysis. By utilizing long-term precipitation and water-sediment data, this study examines the phase differentiation characteristics of the dominant driving factors behind the flood season’s water-sediment changes in the Huangfuchuan Basin over the past 66 years. The findings confirm that alterations in underlying surface conditions have not affected the established runoff-sediment transport function in the basin. Furthermore, the study reveals the coupling effects of vegetation restoration and human activities on the flood season’s water-sediment changes. The ultimate aim of this research is to provide theoretical foundations and data support for comprehensive soil and water conservation measures in the watershed and the establishment of a scientifically informed water-sediment regulation system.”

  1. Conclusion: Authors should improve this section. Authors should shorten this section with significant conclusions, and revise this section for the better understanding of the topic and its future research.

Response: Thank you for the Conclusion suggested. We've taken your advice completely.we had shorten this section with significant conclusions, and revise this section for the better understanding of the topic and its future research.Please check Line 545-577, page 17.we changed this part to “ (1) The relationship between precipitation, runoff, and sediment transport dur-ing the flood season in the Huangfu River Basin follows a power function. The varia-bility of water and sediment is characterized by a threshold relationship with rainfall as the driving factor. Under the same amount of precipitation during the flood season, both runoff and sediment transport exhibit significantly lower values after an abrupt change compared to before the change. Furthermore, the magnitude of change in run-off and sediment transport after the abrupt change is noticeably smaller.(2) The variations in water and sediment in the Huangfu River Basin result from the coupling effects of multiple factors, including underlying surface conditions and meteorological factors. However, the driving forces of these factors do not exhibit syn-ergistic behavior. Sediment transport during the flood season shows a positive correla-tion with precipitation while exhibiting a negative correlation with vegetation cover-age and soil conservation measures. The relationship between sediment transport and soil conservation measures is stronger compared to vegetation coverage.(3) The dominant driving factors of water and sediment variations exhibit tem-poral variations during different historical periods. Prior to the abrupt change in water and sediment conditions, the leading driving factor was flood season rainfall. However, with the intensification of human activities such as large-scale afforestation and check dam construction, soil conservation measures have become the dominant driving force, surpassing the influence of rainfall changes.(4) Despite significant changes in the underlying surface conditions in the Huangfu River Basin, the functional relationship between sediment transport and runoff during the flood season remains unchanged. In other words, the current under-lying surface conditions in the basin do not exert regular regulatory effects on the run-off-sediment relationship.The processes of runoff generation and sediment transport in a watershed exhibit nonlinearity and uncertainty. They represent energy transformation and material transport processes with interrelationships, states, and characteristics. Therefore, fu-ture research needs to be based on systems theory and adopt more scientific and com-prehensive theoretical approaches. This will help determine the quantitative relation-ships and interactions between the various elements of the hydrological system in the Huangfu River Basin, enabling the exploration of simulation and prediction theories and methods for complex watershed systems under strong human-induced disturbances.”

Reviewer 2 Report

The research investigated the relationship between hydrosedimentary variations and their potential governing factors in the Huangfuchuan Basin. The article is not very innovative, there have been many similar studies, but it is well structured and informative. The specific comments are provided as follows.

1.      The authors offer a comprehensive introduction regarding the study of the basin. However, it might be beneficial to incorporate additional information that the audience may find intriguing. It would be appropriate to include a few sentences in the introduction briefly highlighting previous studies that utilize statistical methods to examine the relationship between hydrological characteristics and human activities in the surface water system.

2.      Data and methods: Authors used some well-established methods for the data processing and statistical analysis but please provide some references for the methods the authors used in the study.

3.      Table 6. The significance level is normally provided with the correlation matrix, which is important to identify the correlation between factors with significance.

4.      Section 3.2.2. This part used linear regression here to find the relationship between sediment transport and other factors but I think it is not necessary to be a separate section because it is quite similar results with the correlation analysis.

5.      How were the sample points allocated in the statistical analysis? Because some data are daily and some are monthly, how did the authors align these data temporally?

6.      What significance does this research have for reducing soil and water erosion? How does it assist in the formulation of relevant policies? Additionally, the authors should emphasize the novelty and significance of the article more in the abstract, introduction, and conclusion.

Author Response

We thank the reviewers for the time and effort that they have put into reviewing the previous version of the manuscript. Their suggestions have enabled us to improve our work. Based on the instructions provided, we uploaded the file of the revised manuscript. Accordingly, we have uploaded a copy of the original manuscript with all the changes highlighted by using the track changes mode in MS Word. Appended to this letter is our point-by-point response to the comments raised by the reviewers. The comments are reproduced and our responses are given directly afterward in a different color (red).

Responses to the comments of Reviewer 2

  1. The authors offer a comprehensive introduction regarding the study of the basin. However, it might be beneficial to incorporate additional information that the audience may find intriguing. It would be appropriate to include a few sentences in the introduction briefly highlighting previous studies that utilize statistical methods to examine the relationship between hydrological characteristics and human activities in the surface water system.

Response:We thank the reviewer for the very interesting and very helpful comment. We've taken your advice completely.We had added some sentences to briefly highlighting previous studies that utilize statistical methods to examine the relationship between hydrological characteristics and human activities in the surface water system.Please check Line 81-89, page 1.we changed this part to “The majority of current research on the attribution analysis of water-sediment variations in watersheds primarily focuses on climate change and human activities. The analytical methods employed include hydrological modeling, the dual cumulative curve method, the cumulative slope method, and the elasticity coefficient method, among others. Currently, the investigation of the causal factors underlying hydrological pro-cesses within watersheds is still in its nascent stage, with the mechanisms and syner-gistic effects of influencing factors yet to be clearly elucidated. Furthermore, disentan-gling the specific roles of individual factors from the multitude of coacting elements remains a formidable challenge. ”

  1.  Data and methods: Authors used some well-established methods for the data processing and statistical analysis but please provide some references for the methods the authors used in the study.

Response: We thank the reviewer for the very interesting and very helpful comment.We've taken your advice completely. We had provided some references for the methods the authors used in the study.Please check Line 205-208, page 5.The references is “36. Sang, Y.-F.; Wang, Z.; Liu, C. Comparison of the MK Test and EMD Method for Trend Identification in Hydrological Time Series. Journal of Hydrology 2014, 510, 293–298, doi:10.1016/j.jhydrol.2013.12.039.  37. Kumar, P.; Chandniha, S.K.; Lohani, A.K.; Nema, A.K.; Krishan, G. TREND ANALYSIS OF GROUNDWATER LEVEL USING NON-PARAMETRIC TESTS IN ALLUVIAL AQUIFERS OF UTTAR PRADESH, INDIA. Curr. World Environ 2018, 13, 44–54, doi:10.12944/CWE.13.1.05.  38. Xie, H.; Li, D.; Xiong, L. Exploring the Ability of the Pettitt Method for Detecting Change Point by Monte Carlo Simulation. Stoch Environ Res Risk Assess 2014, 28, 1643–1655, doi:10.1007/s00477-013-0814-y.   39. Lei,C.; Mao,Y. Random Forest Algorithm Based on PCA and Hierarchical Selection Under Spark. Computer Engineering and Applications 2022, 58, 118–127.”

  1. Table 6. The significance level is normally provided with the correlation matrix, which is important to identify the correlation between factors with significance.

Response:We thank the reviewer for the very interesting and very helpful comment.We've taken your advice completely.We had provided the significance level.Please check Line 439-440, page 13.The symbols *** and ** represent the significance levels of 1% and 5%, respectively.

  1. Section 3.2.2. This part used linear regression here to find the relationship between sediment transport and other factors but I think it is not necessary to be a separate section because it is quite similar results with the correlation analysis.

Response:We thank the reviewer for the very interesting and very helpful comment.We've taken your advice completely.We deleted the heading of Section 3.2.2 and Section 3.2.1.

  1. How were the sample points allocated in the statistical analysis? Because some data are daily and some are monthly, how did the authors align these data temporally?

Response: Thank you for the excellent questions raised by the reviewer, which are also key considerations we focused on during data processing and analysis. To facilitate analysis, we converted the daily and monthly data obtained through monitoring mentioned in the paper into annual data by multiplying them by the corresponding number of days or months. To maintain the conciseness of the paper, we did not list this data processing procedure in the paper.

  1. What significance does this research have for reducing soil and water erosion? How does it assist in the formulation of relevant policies? Additionally, the authors should emphasize the novelty and significance of the article more in the abstract, introduction, and conclusion.

Response:Thank you for the excellent questions raised by the reviewer.As mentioned in our paper, the runoff and sediment yield in the Huangfuchuan Basin, located in the hilly and gully area of the Loess Plateau, mainly occur during the flood season. The rainfall from June to September can account for over 80% of the total rainfall during the flood season, and the sediment generated during the flood period can contribute to more than 85% of the annual sediment yield. Therefore, the flood season is an important node for water and soil erosion control in this basin. Our research results indicate that the current vegetation restoration level in the basin is relatively low, making it difficult to effectively control sediment yield during the flood season. Human activities, primarily engineering measures, have played a more significant role in water and soil erosion control in this basin. This research conclusion can provide a reference for local governments and soil conservation policy-making institutions to increase investment in water and soil conservation engineering measures in the basin in the future, indirectly contributing to the reduction of water and soil erosion in the basin. To further emphasize the novelty and significance of the paper, corresponding modifications have been made to the abstract, introduction, and conclusion. Please refer to Line 15-38, Line 101-125, Line 545-577,for specific details.

Abstract:

The Huangfuchuan Basin is one of the major sources of coarse sediment in the Yellow River and has long been a focal point and challenge for the conservation of soil and water in the Yellow River basin. In this study, we analyzed the phase differentiation characteristics of water-sediment variations during the flood season in the Huangfuchuan Basin using a long-term dataset. We elucidated the complex response relationship between water-sediment variations and meteorological factors and human activities, which is of great significance for revealing the mechanisms of water-sediment variations in the region and establishing a scientific water-sediment regulation system in the basin. Statistical methods such as the Mann–Kendall trend test, Sen’s slope estimation, Pettitt nonparametric test, and principal component analysis were employed to identify and analyze the trends and dominant driving factors before and after the water-sediment variations and abrupt changes in parameters such as rainfall and temperature in the Huangfuchuan Basin. Additionally, multiple regression analysis was used to determine the extent of the contribution of climate and human activities to water-sediment variations in the Huangfuchuan Basin. The study revealed that the year 2000 was a turning point for water-sediment variations, with decreases of 11.3%, 76.7%, and 85.1% in flood season rainfall, flood season runoff, and flood season sediment transport, respectively. Despite significant changes in the underlying surface conditions of the Huangfuchuan Basin, the relationship between flood season sediment transport and flood season runoff remained a power-law relationship. In the absence of obvious abrupt changes in temperature, rainfall, and other meteorological factors, the changes in the underlying surface caused by human activities are the main cause of the changes in runoff and sediment yield in the Huangfuchuan Basin. The current level of vegetation restoration in the Huangfuchuan basin is still relatively low, making it difficult to exert stronger control on sediment yield during the flood season, Meanwhile, human activities, primarily based on engineering measures, play a more significant role in the control of soil and water loss in the basin.

Introduction:

Currently, most studies have focused on the analysis of the water-sediment relationship using the annual water and sediment data of the Huangfuchuan Basin. However, it should be noted that the Huangfuchuan Basin, located in the loess hilly and gully region, experiences the majority of its runoff and sediment production during the flood season, which typically occurs from June to September. During this period, rainfall can account for over 80% of the entire flood season’s precipitation, with the sediment generated during the flood period constituting more than 85% of the annual sediment load [32]. Therefore, the flood season serves as a crucial juncture for the management of soil erosion in this basin.

Simultaneously, it is imperative to recognize that the variations in water and sediment are influenced by multiple factors, making the analysis of their dynamics highly intricate. To gain a comprehensive understanding of the water-sediment relationship and the mechanisms driving their changes, it is necessary to eliminate the impacts of nonflood phases. In light of this, the present study employs various analytical methods, including the Mann–Kendall trend test, Sen’s slope estimation, Pettitt nonparametric test, and principal component analysis. By utilizing long-term precipitation and water-sediment data, this study examines the phase differentiation characteristics of the dominant driving factors behind the flood season’s water-sediment changes in the Huangfuchuan Basin over the past 66 years. The findings confirm that alterations in underlying surface conditions have not affected the established runoff-sediment transport function in the basin. Furthermore, the study reveals the coupling effects of vegetation restoration and human activities on the flood season’s water-sediment changes. The ultimate aim of this research is to provide theoretical foundations and data support for comprehensive soil and water conservation measures in the watershed and the establishment of a scientifically informed water-sediment regulation system.

Conclusions:

 (1) The relationship between precipitation, runoff, and sediment transport during the flood season in the Huangfu River Basin follows a power function. The variability of water and sediment is characterized by a threshold relationship with rainfall as the driving factor. Under the same amount of precipitation during the flood season, both runoff and sediment transport exhibit significantly lower values after an abrupt change compared to before the change. Furthermore, the magnitude of change in runoff and sediment transport after the abrupt change is noticeably smaller.

(2) The variations in water and sediment in the Huangfu River Basin result from the coupling effects of multiple factors, including underlying surface conditions and meteorological factors. However, the driving forces of these factors do not exhibit synergistic behavior. Sediment transport during the flood season shows a positive correlation with precipitation while exhibiting a negative correlation with vegetation coverage and soil conservation measures. The relationship between sediment transport and soil conservation measures is stronger compared to vegetation coverage.

(3) The dominant driving factors of water and sediment variations exhibit temporal variations during different historical periods. Prior to the abrupt change in water and sediment conditions, the leading driving factor was flood season rainfall. However, with the intensification of human activities such as large-scale afforestation and check dam construction, soil conservation measures have become the dominant driving force, surpassing the influence of rainfall changes.

(4) Despite significant changes in the underlying surface conditions in the Huangfu River Basin, the functional relationship between sediment transport and runoff during the flood season remains unchanged. In other words, the current underlying surface conditions in the basin do not exert regular regulatory effects on the runoff-sediment relationship.

The processes of runoff generation and sediment transport in a watershed exhibit nonlinearity and uncertainty. They represent energy transformation and material transport processes with interrelationships, states, and characteristics. Therefore, future research needs to be based on systems theory and adopt more scientific and comprehensive theoretical approaches. This will help determine the quantitative relationships and interactions between the various elements of the hydrological system in the Huangfu River Basin, enabling the exploration of simulation and prediction theories and methods for complex watershed systems under strong human-induced disturbances.

Reviewer 3 Report

Line 113: Basis for 70 years not given. In Section 2.2.5 the trend analysis using Pettitt test indicates 66 years for studying the water-sediment variations (Lines 201-203). Clarify on the variations in period of reporting

Line 316: Reduction coefficient stated (5.3165 x 10^6) is different from the one in Equation 11 (5.3165x10^-8). Correct or clarify

Lines 324-327: Confirm on the values reported based on the equations referred to

Lines 504-505: Provide a citation to corroborate the reported findings

Quality of English is okay in presenting the paper as a whole

Author Response

We thank the reviewers for the time and effort that they have put into reviewing the previous version of the manuscript. Their suggestions have enabled us to improve our work. Based on the instructions provided, we uploaded the file of the revised manuscript. Accordingly, we have uploaded a copy of the original manuscript with all the changes highlighted by using the track changes mode in MS Word. Appended to this letter is our point-by-point response to the comments raised by the reviewers. The comments are reproduced and our responses are given directly afterward in a different color (red).

Responses to the comments of Reviewer 3

      1.Line 113: Basis for 70 years not given. In Section 2.2.5 the trend analysis using Pettitt test indicates 66 years for studying the water-sediment variations (Lines 201-203). Clarify on the variations in period of reporting

Response: We thank the reviewer for the very interesting and very helpful comment.We've taken your advice completely. The correct expression should be “66 years” and we have already made the correction within the paper.

       2.Line 316: Reduction coefficient stated (5.3165 x 10^6) is different from the one in Equation 11 (5.3165x10^-8). Correct or clarify

Response: We thank the reviewer for the very interesting and very helpful comment.We've taken your advice completely. The correct expression should be 5.3165 x 10^-6 and we have already made the correction within the paper.

       3. Lines 324-327: Confirm on the values reported based on the equations referred to

Response:We thank the reviewer for the very helpful comment.we have already confirmed on the values reported based on the equations referred to. Please check Line 356-359.we changed this part to “For example, at a flood season precipitation of 370 mm, the runoff before the mutation was 220.11 million m3, whereas after the mutation, it was reduced to 55.33 million m3, representing a decrease of 74.9%. ”

      4.Lines 504-505: Provide a citation to corroborate the reported findings

Response:We thank the reviewer for the very helpful comment.we have already added the a citation to corroborate the reported findings at Line 536: Yao, W., Ran, D., Chen, J. Recent changes in runoff and sediment regimes and future projections in the Yellow River basin. Advances in Water Science 2013, 24, 607–616, doi:10.14042/j.cnki.32.1309.2013.05.013. In the conclusion section of Yao’s article, it is explicitly stated that the functional relationship between annual runoff and sediment yield, as well as the relationship between flood discharge and sediment yield, in most tributaries of the middle reaches of the Yellow River has not changed. In other words, the sediment transport per unit runoff has not decreased, but the synchronous decrease of runoff and sediment yield has occurred.

Round 2

Reviewer 2 Report

The quality of the manuscript has been well improved after the revision.